# Intratumoral core microbiota predicts prognosis and therapeutic response in gastrointestinal cancers

Jia Liu,[1] Dongsheng Wei,[1] Yue Chen,[2] Xingzhong Liu[1]

**ABSTRACT** Microbial communities residing in tumors constitute a critical component of the tumor microenvironment, particularly in gastrointestinal cancers arising from mucosal sites. However, the relationship between microbiota and oncogenesis, as well as its clinical applications, remains underexplored. Here, we performed a comprehensive analysis of the tumor microbiome across six gastrointestinal cancer types and identified a core microbiota composed of 15 bacterial genera associated with patient prognosis. Using bacterial abundance data, we developed a microbiota-based prognostic model to calculate a "risk score." Patients with high-risk scores exhibited poorer prognosis and increased metastatic potential, driven by the activation of tumor metastasis-related signaling pathways, including epithelial-mesenchymal transition, angiogenesis, KRAS, and TGFβ signaling. Furthermore, the model predicted sensitivity to anti-cancer drugs, identifying XL999 and tandutinib as potential targeted therapies for high-risk patients. The core microbiota is also linked to host immunity; patients with high-risk scores were less likely to benefit from immunotherapy. Taken together, these findings highlight the potential of a microbiota-based prognostic model to enhance cancer diagnostics, inform therapeutic decision-making, and advance personalized medicine for gastrointestinal cancers.

**IMPORTANCE** Intratumoral microbiota influence cancer progression, yet their prognostic potential remains underutilized. This study identifies a core microbiota of 15 bacterial genera associated with survival in gastrointestinal cancers and develops a microbiota-based prognostic model. Unlike traditional gene-based models, this approach stratifies patients by microbial signatures, linking high-risk scores to enhanced metastasis via epithelial-mesenchymal transition, angiogenesis, and KRAS signaling. Additionally, we identify XL999 and tandutinib as potential therapies for high-risk patients and reveal that microbiota composition correlates with immunotherapy response. By integrating microbiome profiling into cancer prognosis and treatment selection, this study offers a novel strategy for precision oncology, advancing microbial biomarkers for risk assessment, drug selection, and personalized immunotherapy.

**KEYWORDS** tumor microbiome, TME, tumorigenesis, metastasis, drug response, immunotherapy

Microbial communities inhabit diverse niches within the human body, including the gut, skin, and even tumor tissues, where they play crucial roles in modulating the host immune system and influencing responses to therapies (1–4). Recent advancements in detecting technologies and in-depth exploration of the tumor microbiome have revealed differences in microbiota composition and abundance between tumor and normal tissues (5–7). Special microbiota are enriched in certain tumors compared to normal tissues, such as *Bacteroidetes*, *Firmicutes*, and *Fusobacteria* in esophageal adenocarcinoma (8, 9); *Chlamydia*, *Mycoplasma*, *Acinetobacter*, and *Brucella* in ovarian

Address correspondence to Yue Chen, yuechen@tust.edu.cn, or Xingzhong Liu, liuxz@nankai.edu.cn.

The authors declare no conflicts of interest.

See the funding table on p. 13.

cancers (10); *Atopobium* sp. and *Porphyromonas* sp. in endometrial cancers (11); and *Gammaproteobacteria* in pancreatic cancers (12). These distinct microbiome signatures across various cancers suggest their potential as diagnosis and prognosis indicators, complementing traditional gene-based prognostic models (13).

Intratumoral microorganisms have a significant influence on tumor metastasis, drug efficiency, and host immunity. For instance, *Staphylococcus*, *Lactobacillus*, and *Enterococcus* promote breast cancer metastasis (14); *Escherichia coli* metabolizes the anti-cancer drug 5-fluorouracil to protect colon cancer cells (15); and *Fusobacterium nucleatum* exerts an immunosuppressive effect to shield esophageal and colon cancer tumor tissues from immune system attacks (16). These findings suggest that intratumoral microorganisms could serve as therapeutic targets, particularly in highly heterogeneous cancers like advanced gastrointestinal cancers, where traditional chemotherapy is often ineffective (17). Consequently, a prognosis model based on intratumoral microbiota presents a novel approach for precision cancer therapy, enhancing cancer diagnosis, patient stratification, and treatment response prediction.

Gastrointestinal cancers account for approximately one-quarter of the global cancer incidence and one-third of cancer-related deaths (18). The survival rate for patients with gastrointestinal tumors is closely linked to early-stage diagnosis, yet current diagnostic methods lack sufficient efficiency and accuracy (19–22). The gastrointestinal tract, with its extensive mucosal surface area, provides an ideal habitat for microorganisms, which are more abundant in gastrointestinal tumors than in other cancers (23, 24). This high microbial abundance presents an opportunity to develop microbial-based prognostic models.

Here, we identified a core microbiota composed of 15 bacterial genera closely associated with patient survival risk and developed a microbiota-based prognostic model based on bacterial abundance to calculate the risk score to predict outcomes in gastrointestinal tumors. The high-risk scores exhibited enhanced tumor metastasis, increased sensitivity to two non-gastrointestinal anti-cancer drugs, XL999 and tandutinib, and poor response to immunotherapy. These novel intratumoral microbiota-based prognostic signatures offer promising applications for gastrointestinal cancer diagnostics, drug selection, and personal immunotherapy.

## RESULTS

### Construction of a microbiota-based prognostic model for gastrointestinal cancers

To identify microbiota signatures associated with tumorigenesis, microbiota profiles were analyzed across six types of gastrointestinal cancers (cholangiocarcinoma, colon, esophageal, liver hepatocellular, pancreatic, and stomach adenocarcinoma) and their adjacent normal tissues (Fig. 1a). A total of 1,163 bacterial abundances were recorded for each sample at the genus level (Fig. 1b and c). Univariate Cox regression analysis (FDR < 0.05) identified 52 genera significantly associated with patient prognosis. Least absolute shrinkage and selection operator (LASSO) regression analysis was then applied to reduce the number of candidate genera, and 35 genera were selected (Fig. 1d and e). Multivariate Cox regression analysis (FDR < 0.05) subsequently identified a core microbiota composed of 15 genera—14 high-risk and 1 low-risk-serving as independent prognostic factors (Fig. 1f). These 15 bacterial genera were predominantly enriched in tumor tissues, with lower abundance in normal tissues (Fig. S1). These results suggest that these 15 genera could serve as potential diagnostic markers for patients with gastrointestinal tumors. An intratumoral microbiota-based prognostic model was then constructed based on the risk coefficients of each genus and displayed as the following equation: Risk score = $(-0.135) \times$ Abundance $_{Dorea}$ + $0.930 \times$ Abundance $_{Cosenzaea}$ + $0.800 \times$ Abundance $_{Thioalkalispira}$ + $0.067 \times$ Abundance $_{Granulicella}$ + $0.513 \times$ Abundance $_{Syntrophococcus}$ + $1.392 \times$ Abundance $_{Catenuloplanes}$ + $0.308 \times$ Abundance $_{Turicella}$ + $0.382 \times$ Abundance $_{Formosa}$ + $0.157 \times$ Abundance $_{Candidatus\ Carsonella}$ + $0.572 \times$ Abundance

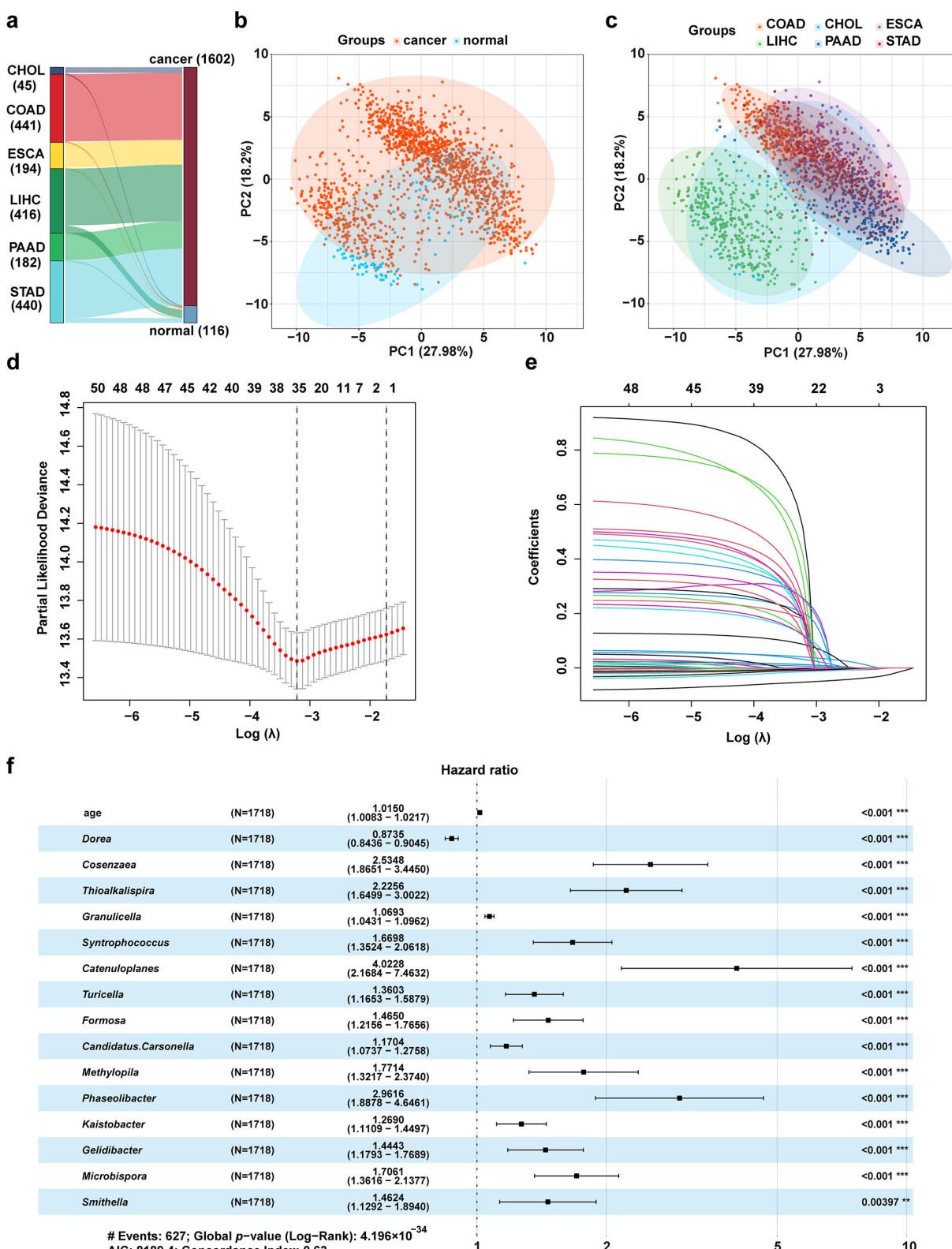

FIG 1 Construction of a microbiota-based prognostic model for gastrointestinal cancers. (a) Type and number of human samples from the TCGA database analyzed in the study. (b and c) Principal component analysis (PCA) of intratumoral microbiome abundance. The first two principal components (PC1 and PC2) and the percentage of variance explained are shown. Each plot represents an individual microbial profile derived from a tumoral sample. (d) The partial likelihood deviance for the LASSO Cox regression analysis. (e) LASSO coefficient plots for each independent variable, calculated using 84 genera

Fig 1 (Continued)

of the intratumoral microbiome. (f) Multivariable Cox analysis facilitated the screening of 16 genera for the construction of the prognostic model. CHOL, cholangiocarcinoma; COAD, colon adenocarcinoma; ESCA, esophageal carcinoma; LIHC, liver hepatocellular carcinoma; PAAD, pancreatic adenocarcinoma; STAD, stomach adenocarcinoma. * $P < 0.05$; ** $P < 0.01$; *** $P < 0.001$.

$Methylopila$ + 1.086 × Abundance $_{Phaseolibacter}$ + 0.238 × Abundance $_{Kaistobacter}$ + 0.368 × Abundance $_{Gelidibacter}$ + 0.534 × Abundance $_{Microbispora}$ + 0.380 × Abundance $_{Smithella}$.

## Evaluation of the predictive efficiency of the risk model for prognosis

To evaluate the predictive efficacy of the microbiota-based prognostic model, the gastrointestinal cancer cohort was used as a training data set. Risk scores were calculated and plotted for each patient (Fig. 2a). Patients with high-risk scores ($n = 187$) exhibited a significantly higher mortality risk compared to those with low-risk scores ($n = 1,531$) (Fig. 2a). Survival analysis further confirmed a poorer prognosis for high-risk patients (Fig. 2b). The area under the receiver operating characteristic (ROC) curve for 1-, 3-, and 5-year survival exceeded 0.6 (Fig. 2c), indicating high predictive accuracy. To further validate the predictive performance of our model, six independent validation cohorts —CHOL, COAD, ESCA, LIHC, PAAD, and STAD—were tested. High-risk groups in these validation cohorts consistently showed significantly higher mortality risks and shorter overall survival compared to low-risk groups (Fig. 2d through i), mirroring the findings in the training cohort. These results demonstrated that significant microbiota signatures are associated with prognosis, and the developed model has the potential to precisely predict survival and prognosis in patients with gastrointestinal cancers.

## Association of risk scores predicted by the prognostic model with malignant tumor signaling pathways

To further elucidate the role of microbiota in tumor development and progression, differentially expressed genes (DEGs) between high- and low-risk score groups were analyzed via gene set enrichment analysis (GSEA). The results revealed that the key tumorigenesis signaling pathways, including KRAS and TGF-β signaling, were significantly activated in the group with high-risk scores (Fig. 3a through c). Notably, tumor tissues with high-risk score exhibited a tendency toward metastasis, as evidenced by significant activation of epithelial-mesenchymal transition (EMT), angiogenesis, and coagulation pathways (Fig. 3d through e). Based on these results, we deduced that the microbiota is involved in tumorigenesis and tumor metastasis.

## Microbiota-based prognostic model predicts responses to anti-cancer drugs

Variations in tumor microenvironment (TME) can lead to significant patient responses to anti-cancer drugs, and microbiota in TME may potentially influence drug sensitivity (25). To test this hypothesis, GSEA was performed on differentially expressed drug target genes between high- and low-risk score groups (Fig. 4a). The results revealed that target genes of anti-cancer drugs such as XL999 and tandutinib, which are not currently used in clinical settings for gastrointestinal tumors, were more highly expressed in the high-risk score group (Fig. 4b and c), suggesting their potential as therapeutic options for patients with high-risk scores. In contrast, target genes of common anti-cancer drugs, such as apigenin and lapatinib, were significantly lower expressed in the high-risk group (Fig. 4d and e), indicating these drugs may be less effective in patients with high-risk scores. Additionally, docking analyzes showed that XL999 and tandutinib could bind to SNAI1, a key transcription factor regulating EMT. Both drugs exhibited binding affinities to the SNAI1 protein below 5 kcal/mol, suggesting the targeted interaction further supports those two drugs in the therapeutic application option. These findings highlight the potential of the microbiota-based model to guide drug selection in a clinical setting.

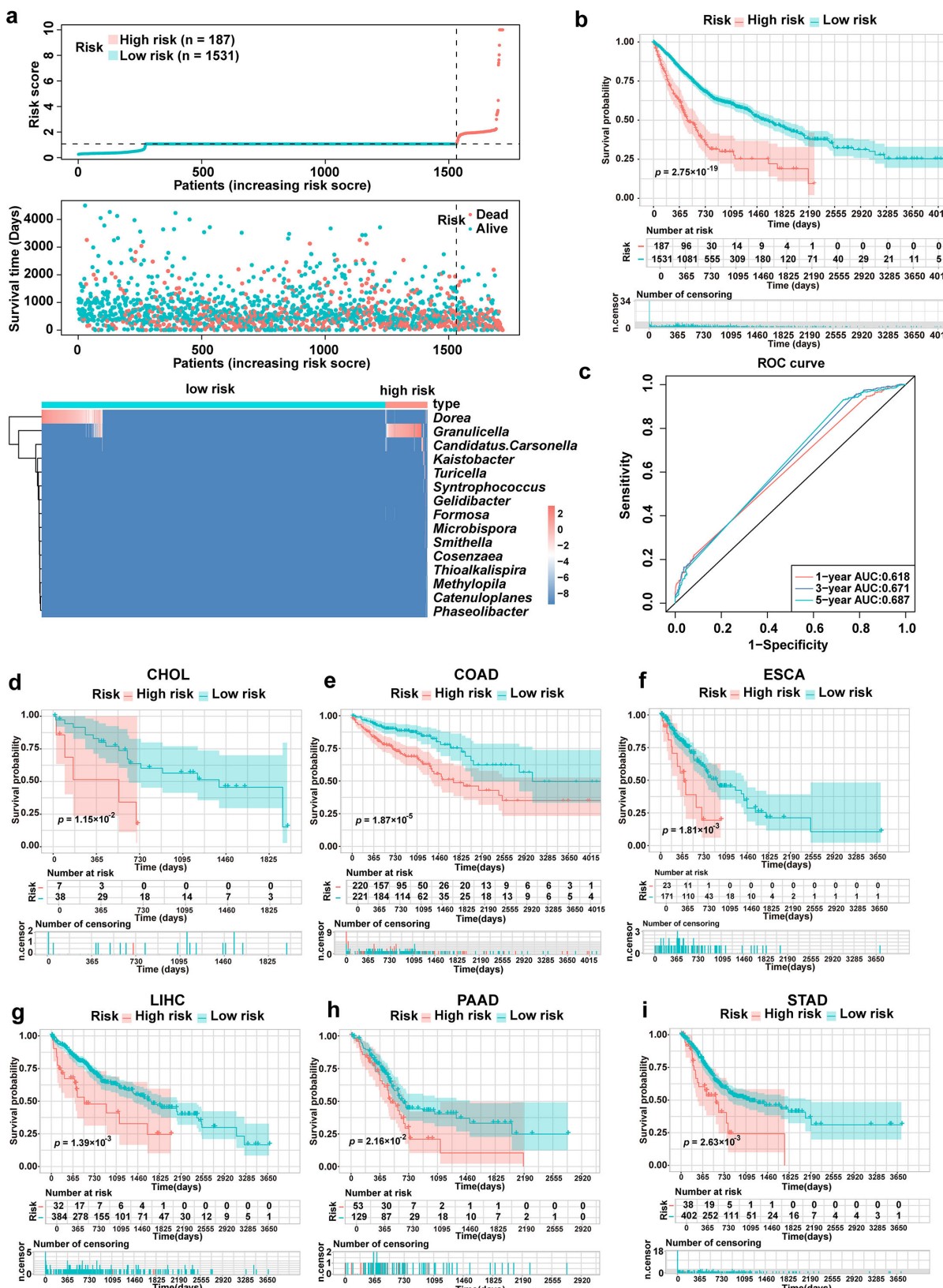

**FIG 2** Evaluation of the predictive efficiency of the risk model for prognosis. (a) Risk score curve, survival status diagram, and bacteria abundance heatmap demonstrate a positive correlation between the model risk score and bacterial abundance of the six-gastrointestinal-cancer training cohort. (b) Kaplan-Meier curve of the 16-genera model for the six-gastrointestinal-cancer training cohort. (c) ROC curve depicting the efficiency of the prognostic model in predicting 1-, 3-, and 5-year overall survival of patients with gastrointestinal cancers. (d–i) Kaplan-Meier curve of the 16-genera model for validation cohort including CHOL (d), COAD (e), ESCA (f), LIHC (g), PAAD (h), and STAD (i).

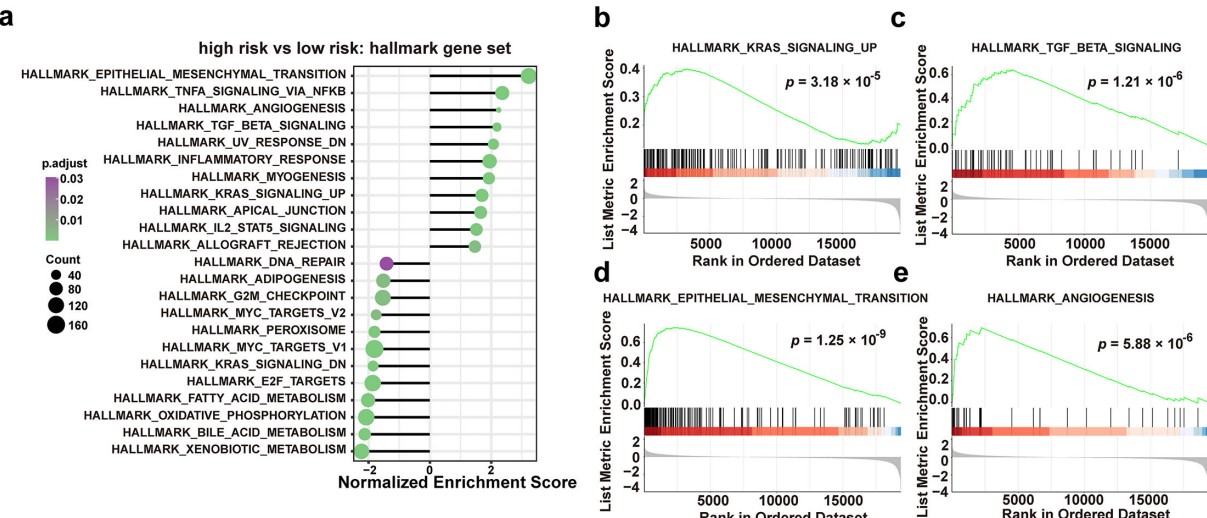

**FIG 3** Association of risk scores predicted by prognostic model with malignant tumor signaling pathways. (a) Gene set enrichment analysis (GSEA) using Hallmark gene sets to identify differentially expressed genes between high- and low-risk patients. (b–e) GSEA of differentially expressed genes related to KRAS signaling (b) and TGF-β signaling (c), epithelial-mesenchymal transition (d), and angiogenesis (e).

## Microbiota-based prognostic model predicts responses to immunotherapy

Based on the single sample GSEA (ssGSEA) analysis, we calculated the abundance of 28 immune cell types in tumor tissues between high- and low-risk score groups. The results showed that the abundance of central memory CD4$^+$ T, effector memory CD8$^+$ T, macrophage, mast, natural killer, natural killer T, and plasmacytoid dendritic were significantly higher ($P < 0.0001$) in the high-risk score group than in the low-risk score group (Fig. 5a). Meanwhile, the abundance of activated CD8$^+$ T, CD56 bright natural killer, γδT, memory B cells, and monocytes was significantly higher ($P < 0.01$) in the low-risk group than in the high-risk group (Fig. 5a through e). These results indicate that the microbiota might influence the tumor immune microenvironment and impact the antitumor immune response.

To further evaluate the response to immunotherapy in high- and low-risk groups, the TIDE value, a combined measure of dysfunction and exclusion values, was applied to predict immune escape potential, and the higher values indicated a lower likelihood of benefiting from immunotherapy. The TIDE, dysfunction, and exclusion values were significantly lower in the low-risk score group compared to the high-risk score group (Fig. 5f through h). Furthermore, microsatellite instability (MSI) values calculated by TIDE software, were significantly lower in the high-risk score group than in the low-risk score group (Fig. 5i), indicating that patients in high-risk score group might have a poor prognosis following immunotherapy. These results demonstrate that the microbiota-based model can predict responses to immunotherapy, and high-risk patients show a poor response.

## Core microbiota response to immune cells in TME

To investigate the interaction between the microbiota from the model and TME, we performed a correlation analysis between the abundance of 15 bacterial genera, 28 types of tumor-infiltrating immune cells, and 50 typical biological process signaling pathways. The analyses revealed that *Dorea* and *Granulicella* were significantly correlated with immune cells and biological pathways (Fig. 6a). Specifically, *Dorea* was positively correlated with activated CD8$^+$ T cells, while *Granulicella* exhibited a negative correlation with activated CD8$^+$ T cells (Fig. 6b and c). Additionally, *Granulicella* was significantly positively correlated with tumor metastasis signaling pathways, whereas *Dorea* showed the opposite trend (Fig. S2; Fig. 6b and c), suggesting that *Granulicella* is associated

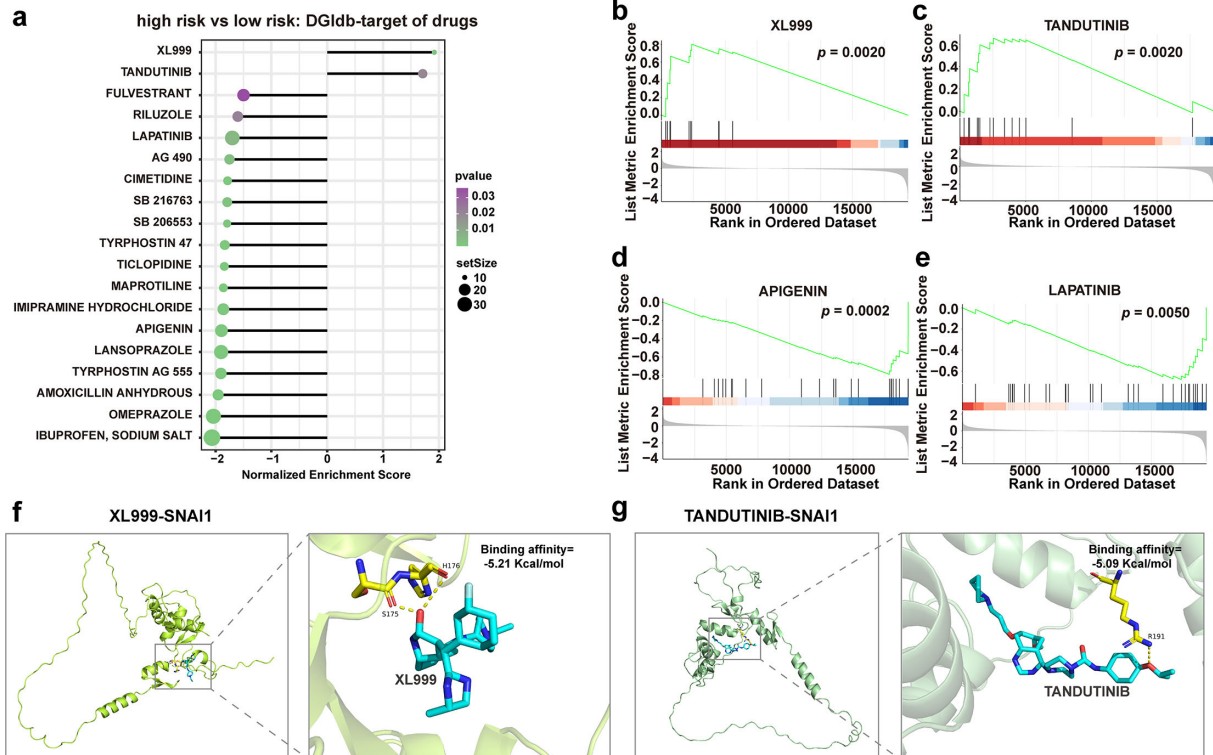

**FIG 4** Microbiota-based prognostic model predicts responses to anti-cancer drugs. (a) GSEA using drug-gene interaction database (DGIdb) to identify differential genes between high- and low-risk patients. (b–e) GSEA of differential genes related to the drugs XL999 (b), tandutinib (c), apigenin, (d) and lapatinib (e). (f and g) Molecular docking analysis of SNAI1 and XL999 or Tandutinib.

with tumor metastasis. Pearson correlation analysis indicated that the risk score was positively correlated with the abundance of *Granulicella*, but negatively correlated with the abundance of *Dorea* (Fig. 6d and e). These results suggest that *Granulicella* may contribute to the poorer immunotherapy response observed in high-risk patients, as its abundance is closely associated with reduced CD8$^+$ T cell infiltration in tumor tissue and a higher risk of metastasis.

## DISCUSSION

Clinical diagnosis and prognosis primarily rely on whole-genome and transcriptome sequencing of tumor tissues to construct prognostic models based on marker genes (26, 27). However, few models incorporate the intratumoral microbiota. The intratumoral microbiota exhibits highly heterogeneous composition and specific abundance across different tumors, playing a unique role in tumor initiation and progression. This suggests that intratumoral microbiota may serve as signatures and indicators of tumors and their progress (7, 28, 29). Emerging bioinformatics analysis methods can capture microbial reads from existing transcriptome sequencing data (30–33). For example, sRNAnalyzer identifies and screens microbiome data from small-RNA sequencing data in TCGA database (34), providing the basis for the development of a microbiota-based prognostic model. Intratumoral microbiota-based prognostic model has been developed for bladder cancer (35); however, such a model remains scarce for gastrointestinal cancers. In this study, we identified 15 bacterial genera as core microbiota most strongly associated with the survival risk of patients with gastrointestinal cancers. A microbiota-based prognostic model built on this core microbiota shows promising potential for enhancing gastrointestinal cancer diagnostics, drug selection, and personalized immunotherapy. However, we acknowledge that the validation cohorts used in Fig. 2d through i were derived from different gastrointestinal cancer types within the same TCGA/BIC data set and not from

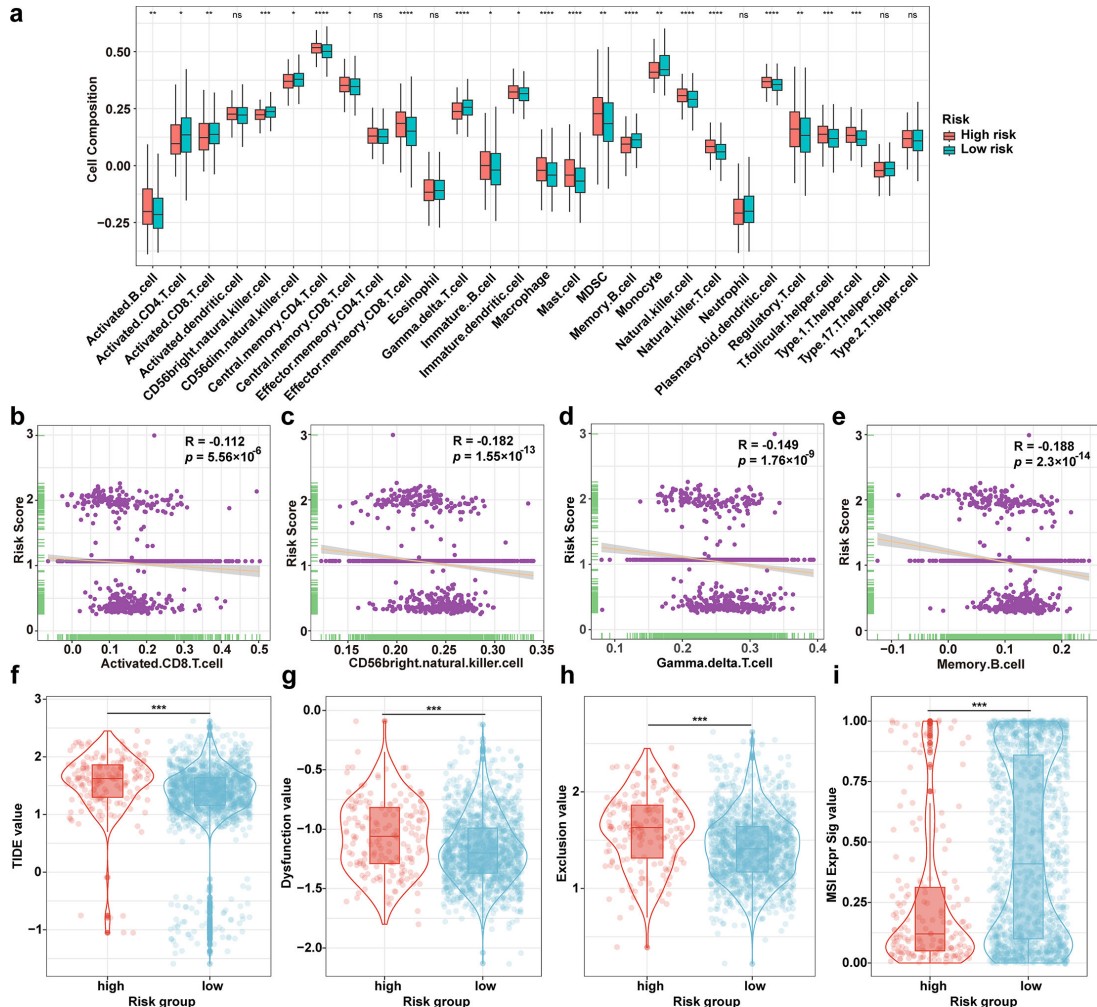

**FIG 5** Microbiota-based prognostic model predicts responses to immunotherapy. (a) The abundance of the immune cells in tumor tissues from high- and low-risk groups. (b–e) Correlation analysis of immune cell abundance and risk score. (f–i) Analysis of the difference in TIDE scores between high- and low-risk scores. *$P < 0.05$; **$P < 0.01$; ***$P < 0.001$; and ****$P < 0.0001$; ns, not significant.

independent external sources. Although these cancer types were analyzed separately from the training cohort, this does not constitute true external validation and may carry risks of circularity or shared biases. To our knowledge, no GEO data sets currently provide sufficiently comprehensive intratumoral microbiota profiles for gastrointestinal cancers. We have therefore highlighted this limitation and emphasized the need for future validation using independent or prospectively collected microbiome data sets to better assess the generalizability of our model.

Compared to gene-based models, microbiota-based prognostic models have their own advantages in guiding diagnosis and treatment. The composition of the microbiome is highly dynamic, influenced by factors such as environment, diet, and lifestyle. This adaptability allows microbiota-based models to more effectively capture a patient's current status and potential health risks, providing a more personalized and real-time assessment (36). Additionally, the clinical application of microbiota-based models is user-friendly and straightforward (36). During gastrointestinal endoscopy, tissue samples from polyps or malignant tumors can be collected for microbiome sequencing. Patient risk scores are then calculated based on the abundance of core microbiota, enabling the assessment of prognosis, metastasis risk, and providing guidance for medication and immunotherapy. However, it is important to acknowledge that microbiota-based models alone exhibit only modest predictive power, as reflected in our AUC values ranging from

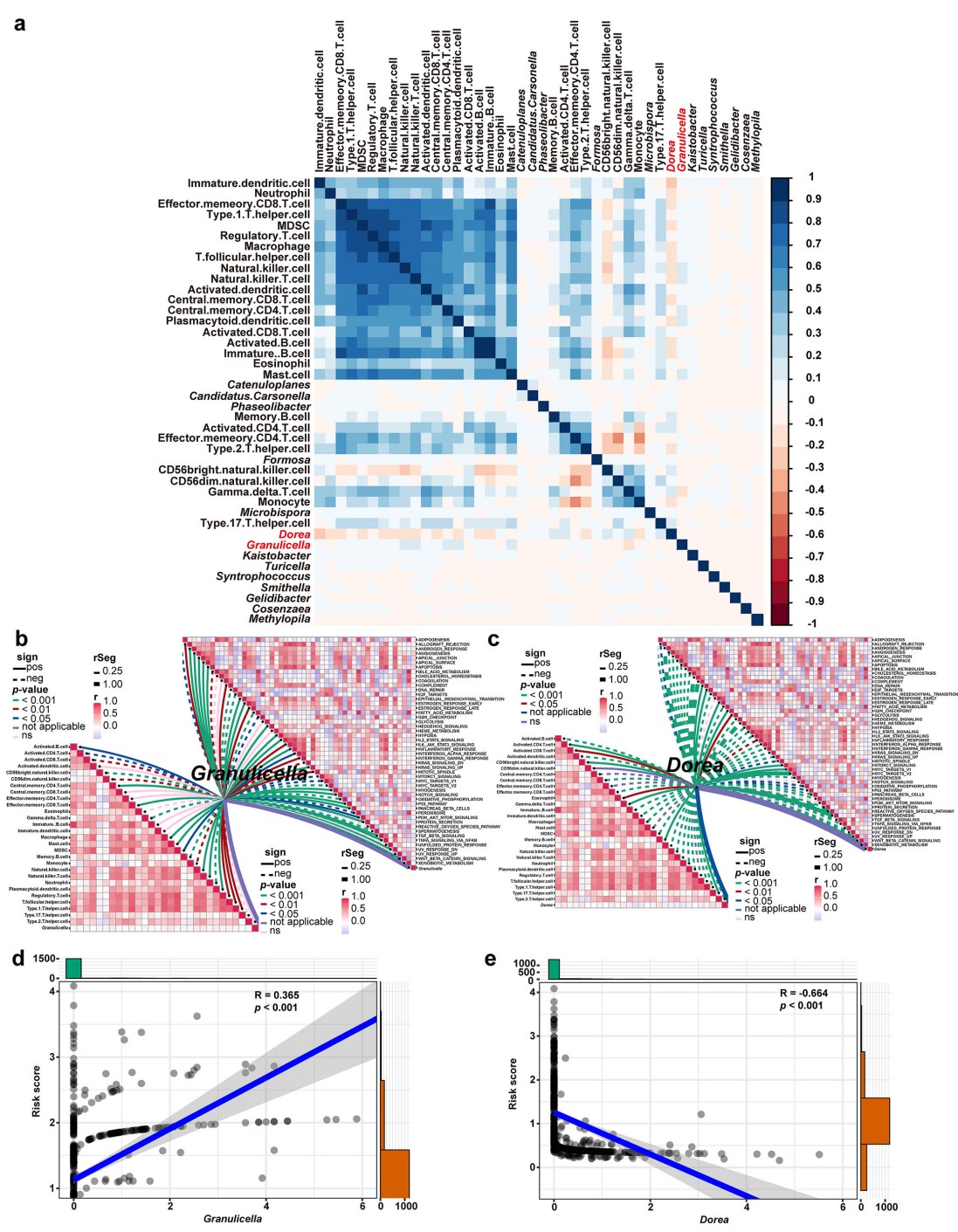

**FIG 6** Core microbiota response to immune cells in TME. (a) Heatmap showing the correlation between the abundance of the 16 genera and the scores of 28 immune cell infiltrations. (b) Correlations between *Granulicella* and scores of the28 immune cell infiltrations or enrichment scores of immunotherapy-predicted pathways. (c) Correlations between *Dorea* and the scores of 28 immune cell infiltrations or enrichment scores of immunotherapy-predicted pathways. (d and e) Correlation analysis of *Dorea* and *Granulicella* abundance and risk score.

0.6 to 0.7. These models should not be viewed as replacements for established clinical parameters such as TNM staging or genomic biomarkers, but rather as complementary tools. Due to data limitations, our study did not include such clinical variables, but future research should aim to integrate microbiome profiles with genomic and clinical features to construct multi-dimensional models with enhanced prognostic utility.

Intratumoral microbiota may influence tumor progression and patient survival (37). The model established in this study has demonstrated that patients with high-risk scores exhibited poorer prognosis and increased metastatic potential. The high-risk score group exhibited activated cancer-promoting signaling pathways, including KRAS and TGF-β. Additionally, the microbiota was strongly associated with *KRAS* and *SMAD4* mutations within these pathways. The high-risk score also correlated with enhanced metastasis-related signaling pathways, such as EMT and angiogenesis. Previous studies have shown that mutations in *KRAS* and *SMAD4* drive epithelial mesenchymal transformation and fibrosis-related gene expression through *RREB1*-mediated enhancer activation, promoting metastasis in lung adenocarcinoma (38). Based on these findings, we deduce that the core microbiota may contribute to tumor metastasis by inducing DNA damage, which promotes *KRAS* and *SMAD4* mutations in tumor cells, thereby activating the EMT pathway. However, experimental validation is needed to confirm the hypothesis.

Intratumoral bacteria influence the efficacy of cancer treatments by modulating drug metabolism (39), highlighting that analyzing the composition of intratumoral microorganisms may serve as a predictor of drug response. In our analysis, patients in the high-risk group showed resistance to certain clinically used anti-cancer drugs, such as apigenin and lapatinib, while demonstrating increased sensitivity to non-gastrointestinal agents, including XL999 and tandutinib. Additionally, the tumor microbiota may modulate the immune microenvironment and influence immunotherapy response (24, 40), although the underlying mechanisms remain largely unexplored. Our findings indicate that patients in the high-risk group exhibit reduced CD8$^+$ T cell infiltration and poorer response to immunotherapy. Notably, the genus *Granulicella* was more abundant in this group and was correlated with both unfavorable prognosis and decreased CD8$^+$ T cell presence. While these associations are intriguing, we emphasize that our study is observational and does not establish causality. Future mechanistic studies—such as those employing germ-free mouse models or organoid systems—will be essential to determine whether *Granulicella* directly modulates immune cell infiltration or metastatic potential. If such effects are confirmed, adjuvant therapies targeting *Granulicella* or its metabolites could offer novel strategies to enhance immunotherapy efficacy in high-risk patients.

The primary limitation of the microbiota-based model lies in its limited applicability to non-gastrointestinal tumors. This is due to the relatively low microbial biomass in the TME, which increases the likelihood of contamination by excessive host DNA, thereby compromising the accuracy of microbiome data (41). Additionally, sample collection for non-gastrointestinal tumors often requires complex surgical procedures, imposing a greater burden on patients. Future prognostic models could integrate microbiome abundance with host gene expression data to enhance accuracy. Beyond tumor diseases, microbiota-based prognostic models also hold great potential in the treatment of non-tumor diseases. Gut microbiota is associated with various conditions, including colitis, obesity, and type 2 diabetes (42–44). Prognostic models leveraging intestinal flora could guide clinical diagnosis and treatment in these conditions as well.

In conclusion, we identified a core microbiota consisting of 15 bacterial genera, which can be used to develop a prognostic model with the potential to precisely predict outcomes in gastrointestinal cancers. The microbiota-mediated model offers valuable insights for drug selection and immunotherapy, improving clinical decision-making.

## MATERIALS AND METHODS

### Data source

Microbiota profiles for 1,602 tumor tissues and 116 adjacent normal tissues from six gastrointestinal cancer types (CHOL, cholangiocarcinoma; COAD, colon adenocarcinoma; ESCA, esophageal carcinoma; LIHC, liver hepatocellular carcinoma; PAAD, pancreatic adenocarcinoma; STAD, stomach adenocarcinoma) at the genus level were retrieved from the BIC database (http://bic.jhlab.tw/). The corresponding transcriptional profiles

and clinical data were obtained from The Cancer Genome Altas (https://portal.gdc.cancer.gov/).

## Microbiome quality control and contamination filtering

Given the low microbial biomass of tumor tissues and the known risk of contamination in microbiome studies based on TCGA sequencing data, we relied on microbiota profiles curated by the BIC database, which implements rigorous quality control and computational decontamination pipelines, including removal of likely environmental and reagent contaminants, normalization across batches, and application of filtering thresholds to reduce noise (34, 45). These preprocessing steps were guided by best practices in the field, as emphasized by Salter et al. (41) and Dohlman et al. (30), who highlighted the importance of contamination-aware analysis in low-biomass samples. Although contamination cannot be fully ruled out, our downstream analyzes focused on genera reproducibly enriched across cancer types, increasing confidence in their biological relevance.

## LASSO and construction of the prognostic model

Bacterial genera were initially evaluated using univariate Cox regression analysis. Reliable predictors were then selected through LASSO analysis. The risk score for each patient in the TCGA database was calculated using the following formula:

$$\text{Risk score} = \Sigma \left( \text{coefficient}_{\text{genusn}} \times \text{Abundance level}_{\text{genusn}} \right)$$

Subsequent analyzes assessed the correlation between the risk score and the prognosis of the patients.

## Validation of the accuracy of the prognostic model for predicting the prognosis of patients

The survivalROC R package (version 1.0.3.1) was used to assess the time dependence of the prognostic bacterial signature, comprizing 15 genera, for predicting outcomes in six gastrointestinal cancer types. A two-way log-rank test $P < 0.05$ was considered statistically significant for survival analysis. To stratify patients into risk groups, we calculated individual risk scores using the prognostic model by the survival package and then dichotomized patients at the median risk score (1.07). Patients with scores above the median were classified as high-risk ($n = 368$), while those below were classified as low-risk. The survminer package (version 0.4.9) was used to generate the risk curves and survival status plots. The pheatmap R package (version 1.0.12) was applied to generate a risk heatmap displaying genus-level differences between risk groups.

## Functional enrichment analysis

Normalization and differential expression analysis of the transcriptional profiles were performed using the DESeq2 R packages (version 1.36.0). Based on the risk score, samples were categorized into two groups: high- and low-risk groups. Differential expression analysis was conducted between high- and low-risk groups, identifying DEGs using an adjusted $P$ value < 0.05 and |log2FoldChange| > 1. GSEA of DEGs was performed using the clusterProfiler R package (version 4.4.4) (46), utilizing the Hallmark gene sets (https://www.gsea-msigdb.org/gsea/index.jsp) (47, 48).

## Prediction of drug response

Drug-gene interaction data were retrieved from the drug-gene interaction database (DGIdb; https://dgidb.org/) (49). Genes targeted by inhibitory drugs were identified and used to construct drug response gene sets. GSEA of DEGs between high- and low-risk groups was performed using the clusterProfiler R package, utilizing the drug response gene sets.

## Immune cell infiltration analysis in TME

Gene set variation analysis R package (version 1.46.0) was used to estimate pathway activity by transforming the gene expression data matrix into corresponding gene set enrichment scores. To calculate the enrichment score of immune cells in TME, the ssGSEA algorithm was used. The relative proportions of diverse tumor-infiltrating lymphocytes were inferred for each cancer type, based on the immune-related gene expression profile of 28 immune cell types derived from a previous study (50). Spearman correlation tests were applied to assess the associations between the abundance of immune cell populations and the 16 bacterial genera.

## Docking

The 3D structure of the receptor protein was predicted by AlphaFold Server (51) (https://alphafoldserver.com/), and the 3D structure of the ligand drug was downloaded from the PubChem database (https://pubchem.ncbi.nlm.nih.gov/). The files of the receptor and ligand were preprocessed by Pymol software (version 3.0) and converted into PDB files. The Diffdock tool (https://huggingface.co/spaces/reginabarzilaygroup/DiffDock-Web) was used to perform molecular docking on the PDB files of the protein and drug, and Pymol was used for visualization. The molecular binding energy was calculated by the PRODIGY tool (https://rascar.science.uu.nl/prodigy/) (52).

## Tumor immune dysfunction and exclusion (TIDE)

TIDE is a computational framework developed to evaluate immune checkpoint blockade (ICB) response through predicting the tumor immune escape (including T cell dysfunction and T cell exclusion) from the gene expression profiles of cancer samples. The software is freely available online at http://tide.dfci.harvard.edu. Higher TIDE, dysfunction, and exclusion values indicate a higher possibility of immune escape and poor ICB response for gastrointestinal cancer patients. The TIDE website can also calculate the genomic MSI value. The higher the MSI value, the better the prognosis.

## Statistical analyses

All statistical analyzes were conducted using R software (version 4.2.3). Statistical significance was determined using a two-tailed test with $P < 0.05$ considered significant. The Wilcoxon test was applied to compare differences between the low and high ssGSEA score groups. Spearman's correlation analysis was employed to evaluate the relationships between quantitative variables with non-normal distributions. Survival curves were analyzed using the Kaplan-Meier method and the Cox proportional hazards regression model, with differences assessed via the log-rank test. Survival and regression analyzes were performed using the survival (version 3.5.5) and survminer (version 0.4.9) R packages. Survival curves were visualized using the ggplot2 R package (version 3.5.1).

### ACKNOWLEDGMENTS

We thank Prof. Yongxin Liu (Agricultural Genomics Institute at Shenzhen, Chinese Academy of Agricultural Sciences) for advice on the microbiome analysis and Dr. Shipeng Guo (Children's Hospital, Chongqing Medical University) and Dr. Zhougeng Xu (Institute of Plant Physiology and Ecology, Chinese Academy of Sciences) for advice on the analysis of RNA-seq.

This study was supported by grants from the Fundamental Research Funds for Central Universities (030/C02922104) to X.Z.L. and the National Natural Science Foundation of China (32400033) to Y.C.

J.L.: Data collection and analysis; Conceptualization; Methodology; Software; Visualization; Writing—original draft. Y.C.: Conceptualization; Supervision; Writing—review and editing. D.W: Writing— review and editing. X.L.: Supervision; Writing—review and editing.

## AUTHOR AFFILIATIONS

[1]State Key Laboratory of Medicinal Chemical Biology, Key Laboratory of Molecular Microbiology and Technology of the Ministry of Education, Department of Microbiology, Frontiers Science Center for Cell Responses, College of Life Science, Nankai University, Tianjin, China

[2]College of Biotechnology, Tianjin University of Science and Technology, Tianjin, China

## AUTHOR ORCIDs

Jia Liu  http://orcid.org/0000-0001-6881-2791
Yue Chen  http://orcid.org/0000-0002-3203-8869
Xingzhong Liu  http://orcid.org/0000-0002-3224-8604

## FUNDING

| Funder | Grant(s) | Author(s) |
|---|---|---|
| Fundamental Research Funds for Central Universities of China | 030/C02922104 | Xingzhong Liu |
| National Natural Science Foundation of China | 32400033 | Yue Chen |

## AUTHOR CONTRIBUTIONS

Jia Liu, Conceptualization, Formal analysis, Methodology, Software, Visualization, Writing – original draft | Dongsheng Wei, Writing – review and editing | Yue Chen, Conceptualization, Supervision, Writing – review and editing | Xingzhong Liu, Funding acquisition, Supervision, Writing – review and editing

## DATA AVAILABILITY

The microbiota profiles of gastrointestinal cancer samples were obtained from the BIC database (http://bic.jhlab.tw/). The bulk RNA-seq data set and clinical data were obtained from The Cancer Genome Altas (https://portal.gdc.cancer.gov/). The R software (version 4.2.3) utilized in this study are freely and open source. Source code for bulk RNA-seq and scRNA-seq are available at GitHub (https://github.com/liujialeo/paper_tumor_microbiome).

## ETHICS APPROVAL

No animals or humans were involved in this study.

## ADDITIONAL FILES

The following material is available online.

### Supplemental Material

**Supplemental figures (Spectrum00390-25-s0001.docx).** Fig. S1 and S2.
**Graphic abstract (Spectrum00390-25-s0002.tif).** Graphic abstract.

### Open Peer Review

**PEER REVIEW HISTORY (review-history.pdf).** An accounting of the reviewer comments and feedback.

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
