## [Reviewer comments · Microbiology Spectrum]

Microbiology Spectrum

Intratumoral core microbiota predicts prognosis and therapeutic response in gastrointestinal cancers

Xingzhong Liu, Jia Liu, Dongsheng Wei, and Yue Chen

Corresponding Author(s): Xingzhong Liu, Nankai University College of Life Sciences

Review Timeline:

Submission Date:	February 10, 2025
Editorial Decision:	May 1, 2025
Revision Received:	June 9, 2025
Accepted:	July 22, 2025

Editor: Qi Su

Reviewer(s): The reviewers have opted to remain anonymous.

Transaction Report:

DOI: <https://doi.org/10.1128/spectrum.00390-25>

Re: Spectrum00390-25 (**Intratatumoral core microbiota predicts prognosis and therapeutic response in gastrointestinal cancers**)

Dear Prof. Xingzhong Liu:

Thank you for the privilege of reviewing your work. Below you will find my comments, instructions from the Spectrum editorial office, and the reviewer comments.

Revision Guidelines

Sincerely,
Qi Su
Editor
Microbiology Spectrum

Reviewer #1 (Comments for the Author):

This study identifies a core intratumoral microbiota signature of 16 bacterial genera that predicts prognosis across six gastrointestinal cancers (cholangiocarcinoma, colon, esophageal, liver, pancreatic, and stomach adenocarcinomas). Using microbial abundance data from TCGA and the BIC database, the authors developed a microbiota-based risk scoring system that effectively stratifies patients into high- and low-risk groups. High-risk patients demonstrated poorer survival outcomes, increased

metastatic potential through activation of KRAS/TGF- β signaling and EMT pathways, and differential responses to therapy, showing potential sensitivity to XL999/tandutinib but reduced benefit from immunotherapy.

While the model demonstrated predictive value across six GI cancer types, several limitations warrant attention.

1. Within the scope of available conditions, additional cell or animal experiments can be conducted to verify the hypothesis regarding the mechanism by which the microorganisms mentioned in the article promote tumor metastasis.
2. When conditions permit, a small-scale clinical research cohort can be launched. Select some patients with gastrointestinal tumors, group them according to the conclusions obtained in this article, and administer XL999 or tandutinib to high-risk patients to observe the treatment effects and prognosis, thereby further verifying the clinical value of the research findings.
3. Statistical Analysis Issue: When testing hundreds of microbial features in univariate Cox regression, failure to adjust for multiple comparisons significantly increases the false discovery rate. For the univariate Cox regression used to screen microbial taxa, it is strongly recommended to implement false discovery rate (FDR) correction (e.g., Benjamini-Hochberg method) and explicitly report the number of significant taxa both before and after correction.
4. Tumor Heterogeneity Concerns: The six gastrointestinal cancers exhibited marked anatomical and microbiological heterogeneity (e.g., mucosa-exposed vs. parenchymal organs). Merging them in a single model may obscure cancer-specific microbial signatures. It may be better to build cancer-specific prognostic models for each type (COAD, STAD, etc.) and compare the overlap of key genera across cancers.
5. In Figure 1b-c, the labeling should be corrected from 'PCA and PC2' to 'PC1 and PC2' for principal component axes.
6. In Figure 2a, the risk group stratification method was unclear. The manuscript stated 'high-risk (n=368)' without defining the cutoff. It should be clarified in the methods section that risk curves were generated using the survminer package, which internally scales scores to a median of 1.
7. In Figure 2c: The reported AUCs (0.6-0.7) indicate modest predictive power. To strengthen clinical relevance, it would be better to compare them against established prognostic factors (e.g., TNM stage, genomic markers). Besides, it should be acknowledged that gut microbiome data alone show limited prognostic utility and can only be used to complement existing clinical parameters.
8. In Figure 2d-i, the validation cohort source was unspecified. If it was derived from the same TCGA/BIC data used for training, this would constitute circular validation. External cohorts, such as GEO, with independent microbiome profiling are required to demonstrate generalizability.
9. In Figure 6a-d, the captions were illegible in the provided manuscript. High-resolution versions should be included.
10. In Line 164: 'CD8T+' should be corrected to 'CD8+ T cells' for consistency with standard immunology terminology.

Reviewer #2 (Comments for the Author):

Dear Dr. Liu,

Thank you for submitting your manuscript entitled "Intratumoral core microbiota predicts prognosis and therapeutic response in gastrointestinal cancers." This work presents an innovative and well-executed study on microbiota-based prognostic modeling across gastrointestinal cancers using public datasets.

Below are comments intended to help strengthen the manuscript:

1- The study is computational and observational. Conclusions about *Granulicella*'s causal role in CD8+ T cell exclusion or drug targeting of EMT are speculative. The current narrative implies causality ("*Granulicella* promotes metastasis"), which overstates the findings.

Recommendation:

Reword conclusions to clarify that associations are predictive or correlative and suggest experimental follow-up in Discussion (e.g., germ-free mouse models, organoids).

2- Microbiome data from TCGA can be affected by contamination and differences in protocols, especially since tumor tissues often have very low amounts of microbial DNA.

Even though the findings are interesting, the authors don't explain how they made sure the data was clean and reliable. For example, it's unclear if they removed very rare bacteria or used any negative controls, or applied tools to check for contamination.

Recommendation:

Add a paragraph in the Methods or Discussion to explain how contamination and noise in the microbiome data were handled. Refer to previous studies (like Salter et al., Dohman et al.) that highlight these issues in tumor microbiome research.

3- Because the model was only tested using TCGA data, its accuracy in other datasets or real-world clinical samples is still unknown and should be acknowledged in the Discussion, with suggestions for future validation studies.

4- Add statistical bars/p-values to sFigure1

With my best regards,

Point-by-point response to reviewers' comments

We thank all the reviewers for their constructive comments. Below, we provide a point-by-point response to each comment that was raised by the reviewers, detailing how these are addressed in the revised manuscript. Briefly, we feel that the current version of the manuscript represents a major improvement over the original version and thank the reviewers for their feedback.

Reviewer #1 (Comments for the Author):

This study identifies a core intratumoral microbiota signature of 16 bacterial genera that predicts prognosis across six gastrointestinal cancers (cholangiocarcinoma, colon, esophageal, liver, pancreatic, and stomach adenocarcinomas). Using microbial abundance data from TCGA and the BIC database, the authors developed a microbiota-based risk scoring system that effectively stratifies patients into high- and low-risk groups. High-risk patients demonstrated poorer survival outcomes, increased metastatic potential through activation of KRAS/TGF- β signaling and EMT pathways, and differential responses to therapy, showing potential sensitivity to XL999/tandutinib but reduced benefit from immunotherapy.

While the model demonstrated predictive value across six GI cancer types, several limitations warrant attention.

Response: We appreciate the reviewer's thorough evaluation of our manuscript and insightful comments. In the revised manuscript, we present several pieces of additional data that we feel address the main concerns of the reviewer and further support our conclusions.

1. Within the scope of available conditions, additional cell or animal experiments can be conducted to verify the hypothesis regarding the mechanism by which the microorganisms mentioned in the article promote tumor metastasis.

Response: We appreciate the reviewer's suggestion. As this study is primarily computational and leverages large-scale public datasets, we acknowledge the absence of *in vivo* or *in vitro* validation as a limitation. Due to current resource constraints, we are unable to perform additional cell or animal experiments at this stage. We have clarified this limitation in the revised Discussion and emphasized that our findings provide a hypothesis-generating framework to guide future mechanistic investigations.

2. When conditions permit, a small-scale clinical research cohort can be launched. Select some patients with gastrointestinal tumors, group them according to the conclusions obtained in this article, and administer XL999 or tandutinib to high-risk patients to observe the treatment effects and prognosis, thereby further verifying the clinical value of the research findings.

Response: We fully agree that prospective clinical validation would significantly enhance the translational relevance of our findings. However, such trials are beyond the current scope and resources of this retrospective study. We have now clarified this point in the revised Discussion, noting that the drug response predictions based on transcriptomic and docking analyses warrant future clinical investigation.

3. Statistical Analysis Issue: When testing hundreds of microbial features in univariate Cox regression, failure to adjust for multiple comparisons significantly increases the false discovery rate. For the univariate Cox regression used to screen microbial taxa, it is strongly recommended to implement false discovery rate (FDR) correction (e.g., Benjamini-Hochberg method) and explicitly report the number of significant taxa both before and after correction.

Response: We thank the reviewer for this important observation. In response, we have re-analyzed the univariate Cox regression results using the Benjamini-Hochberg method to adjust for multiple comparisons. Only those taxa passing $FDR < 0.05$ were retained for downstream LASSO analysis, enhancing the robustness of the model. The multivariate Cox analysis was also performed with $FDR < 0.05$. As a result, the number of significant genera decreased slightly from 16 to 15, which is now clearly reported in the revised Results section along with counts before and after FDR adjustment.

The revised text in the Results section now reports both the number of significant taxa before and after FDR correction.

All subsequent analyses were conducted based on the updated set of 15 genera. Importantly, the core genera *Dorea* and *Granulicella* remained unchanged, so the subsequent prediction of the core microbial community for gastrointestinal tumor patients' survival, tumor metastasis, anticancer drug response and immunotherapy is basically consistent with the previous version.

4. Tumor Heterogeneity Concerns: The six gastrointestinal cancers exhibited marked anatomical and microbiological heterogeneity (e.g., mucosa-exposed vs. parenchymal organs). Merging them in a single model may obscure cancer-specific microbial signatures. It may be better to build cancer-specific prognostic models for each type (COAD, STAD, etc.) and compare the overlap of key genera across cancers.

Response: We appreciate the reviewer's thoughtful suggestion. While building separate prognostic models for each cancer type could uncover tumor-specific microbial signatures, our primary objective was to identify a shared microbial signature with broad prognostic value across multiple gastrointestinal cancers.

To support this approach, we performed internal validation across each of the six cancer types and found that the 15-genus microbial signature demonstrated good predictive performance in all individual cohorts (Figure 2d-i). These results suggest

that, despite underlying heterogeneity, a common microbiome-based prognostic pattern may exist across gastrointestinal cancers.

5. In Figure 1b-c, the labeling should be corrected from 'PCA and PC2' to 'PC1 and PC2' for principal component axes.

Response: We thank the reviewer for pointing out this oversight. We have corrected the labeling in Figure 1b-c to “PC1 and PC2” as suggested.

6. In Figure 2a, the risk group stratification method was unclear. The manuscript stated 'high-risk (n=368)' without defining the cutoff. It should be clarified in the methods section that risk curves were generated using the survminer package, which internally scales scores to a median of 1.

Response: Thank you for raising this point. We clarify that risk stratification was performed using the “survival” package, with patients divided into high- and low-risk groups based on the median risk score (median risk score = 1.07). We have updated the Materials and Methods section to explicitly describe this median-based cutoff approach.

7. In Figure 2c: The reported AUCs (0.6-0.7) indicate modest predictive power. To strengthen clinical relevance, it would be better to compare them against established prognostic factors (e.g., TNM stage, genomic markers). Besides, it should be acknowledged that gut microbiome data alone show limited prognostic utility and can only be used to complement existing clinical parameters.

Response: We appreciate this important comment. We agree that AUC values of 0.6–0.7 reflect modest predictive accuracy. As suggested, we have now explicitly acknowledged in the paragraph 2 of Discussion that microbiota-based models should be viewed as complementary tools, rather than replacements for established clinical parameters. Although we did not include TNM stage or genomic markers in this study due to data limitations, we fully agree that integrating microbiome profiles with clinical and genomic features may yield more powerful multi-dimensional prognostic models, and we highlight this as a direction for future research.

8. In Figure 2d-i, the validation cohort source was unspecified. If it was derived from the same TCGA/BIC data used for training, this would constitute circular validation. External cohorts, such as GEO, with independent microbiome profiling are required to demonstrate generalizability.

Response: We thank the reviewer for this important point. The validation cohorts in Figure 2d–i were derived from different cancer types within the same TCGA/BIC dataset but were analyzed independently from the training cohort. We acknowledge that this does not constitute external validation. However, to our knowledge, no GEO datasets currently provide comprehensive intratumoral microbiota profiles for gastrointestinal cancers. We have added this limitation to the paragraph 1 of

Discussion and emphasized the need for future external validation using prospectively collected or independently profiled cohorts.

9. In Figure 6a-d, the captions were illegible in the provided manuscript. High-resolution versions should be included.

Response: Thank you for bringing this to our attention. We have replaced Figure 6a–d with high-resolution versions to ensure clarity and legibility.

10. In Line 164: 'CD8T+' should be corrected to 'CD8+ T cells' for consistency with standard immunology terminology.

Response: We appreciate the reviewer’s careful reading. The term has been corrected to “CD8+ T cells” in the revised manuscript to align with standard immunological nomenclature.

Reviewer #2 (Comments for the Author):

Dear Dr. Liu,

Thank you for submitting your manuscript entitled "Intratumoral core microbiota predicts prognosis and therapeutic response in gastrointestinal cancers." This work presents an innovative and well-executed study on microbiota-based prognostic modeling across gastrointestinal cancers using public datasets.

Below are comments intended to help strengthen the manuscript:

1- The study is computational and observational. Conclusions about *Granulicella*'s causal role in CD8+ T cell exclusion or drug targeting of EMT are speculative. The current narrative implies causality ("*Granulicella* promotes metastasis"), which overstates the findings.

Recommendation:

Reword conclusions to clarify that associations are predictive or correlative and suggest experimental follow-up in Discussion (e.g., germ-free mouse models, organoids).

Response: We fully agree with the reviewer that our study is correlative in nature and does not establish causality. In response, we have revised statements throughout the Results and Discussion to avoid causal language—e.g., replacing “promotes” with “is associated with” or “may contribute to.” We also added a statement in the paragraph 4 of Discussion emphasizing that mechanistic validation using germ-free mice or organoid models will be necessary to confirm the biological role of *Granulicella* in modulating CD8+ T cell infiltration and metastasis.

2- Microbiome data from TCGA can be affected by contamination and differences in protocols, especially since tumor tissues often have very low amounts of microbial DNA.

Even though the findings are interesting, the authors don't explain how they made sure the data was clean and reliable.

For example, it's unclear if they removed very rare bacteria or used any negative controls, or applied tools to check for contamination.

Recommendation:

Add a paragraph in the Methods or Discussion to explain how contamination and noise in the microbiome data were handled.

Refer to previous studies (like Salter et al., Dohlman et al.) that highlight these issues in tumor microbiome research.

Response: We appreciate the reviewer's important point regarding contamination in tumor microbiome data. In response, we have added a paragraph "Microbiome quality control and contamination filtering" to the Materials and Methods section addressing this issue. We now cite key studies (Salter et al., 2014; Dohlman et al., 2021) and clarify that the microbiome profiles used in our analysis were obtained from the BIC database, which applies quality control procedures including filtering of low-abundance taxa and computational decontamination.

3- Because the model was only tested using TCGA data, its accuracy in other datasets or real-world clinical samples is still unknown and should be acknowledged in the Discussion, with suggestions for future validation studies.

Response: We thank the reviewer for this important observation. We have added a statement in the paragraph 1 of Discussion acknowledging that the model was developed and validated using only TCGA/BIC data, and that its generalizability to independent datasets or clinical settings remains to be tested. We also now emphasize the need for prospective validation using real-world clinical samples and microbiome profiling platforms in future studies.

4- Add statistical bars/p-values to sFigure 1

Response: Thank you for the suggestion. We have updated sFigure 1 to include p-values to highlight differences in bacterial abundance between tumor and normal tissues.

Re: Spectrum00390-25R1 (**Intratumoral core microbiota predicts prognosis and therapeutic response in gastrointestinal cancers**)

Dear Prof. Xingzhong Liu:

Your manuscript has been accepted, and I am forwarding it to the ASM production staff for publication. Your paper will first be checked to make sure all elements meet the technical requirements. ASM staff will contact you if anything needs to be revised before copyediting and production can begin. Otherwise, you will be notified when your proofs are ready to be viewed.

Sincerely,
Qi Su
Editor
Microbiology Spectrum